# Aesthetic Evaluation of Facial Scars in Patients Undergoing Surgery for Basal Cell Carcinoma: A Prospective Longitudinal Pilot Study and Validation of POSAS 2.0 in the Lithuanian Language

**DOI:** 10.3390/cancers16112091

**Published:** 2024-05-30

**Authors:** Alvija Kučinskaitė, Domantas Stundys, Simona Gervickaitė, Gabrielė Tarutytė, Jūratė Grigaitienė, Janina Tutkuvienė, Ligita Jančorienė

**Affiliations:** 1Faculty of Medicine, Vilnius University, 01513 Vilnius, Lithuania; 2Institute of Clinical Medicine, Faculty of Medicine, Vilnius University, 01513 Vilnius, Lithuania; jurate.grigaitiene@santa.lt (J.G.); ligita.jancoriene@santa.lt (L.J.); 3Department of Anatomy, Histology and Anthropology, Faculty of Medicine, Vilnius University, 03101 Vilnius, Lithuania; simona.gervickaite@mf.vu.lt (S.G.); janina.tutkuviene@mf.vu.lt (J.T.); 4Department of Research and Innovation, Faculty of Medicine, Vilnius University, 03101 Vilnius, Lithuania; gabriele.tarutyte@santa.lt; 5Institute of Applied Mathematics, Faculty of Mathematics and Informatics, Vilnius University, 03225 Vilnius, Lithuania; 6Vilnius University Hospital Santaros Klinikos, 08661 Vilnius, Lithuania

**Keywords:** scar, aesthetic outcomes, basal cell carcinoma, skin cancer, surgery of the face, skin cancer surgery, health-related quality of life

## Abstract

**Simple Summary:**

The quality of life of patients with facial basal cell carcinoma significantly improves after surgery. Nevertheless, it remains inferior to those without the disease. Facial scarring has been identified as a contributing factor to adverse psychosocial changes. In this study, we explore the aesthetic assessment of facial scars within this specific patient group, aiming to uncover potential correlations between the severity of scars and the quality of life. This study comprises two phases as follows: scale validation and pilot with a sample size of 100 patients. The Lithuanian version of the POSAS 2.0 was established after a thorough psychometric evaluation, surpassing acceptable validity thresholds. The pilot phase findings show a notable improvement in scars during the later stages of postoperative recovery, with the initial identification of specific groups that perceive their scars more negatively. Given the observed correlations between the scar assessment and the quality of life, this study highlights the crucial role of addressing the aesthetic satisfaction of patients with surgically treated basal cell carcinoma.

**Abstract:**

Facial basal cell carcinoma (BCC) surgery enhances the quality of life (QoL) but leaves patients with inferior QoL, presumably caused by scarring, emphasizing the need to understand post-surgery aesthetic satisfaction. This study aimed to validate the Lithuanian version of the Patient and Observer Scar Assessment Scale (POSAS) 2.0 and utilise it to identify scar evaluation differences and correlations among POSAS scores and specific aesthetic facial regions, age, gender, surgery types, and short- and long-term QoL. Employing a prospective longitudinal design, 100 patients with facial scars after surgical BCC removal were enrolled. The validation phase confirmed the translated POSAS 2.0 psychometric properties, while the pilot phase used statistical analyses to compare scores among demographic and clinical groups and evaluate correlations between scar assessment and QoL. The findings indicate that the translated Lithuanian version of POSAS 2.0 exhibits good psychometric properties, revealing insights into aesthetic satisfaction with post-surgical facial scars and their impact on QoL. The Lithuanian version of the POSAS 2.0 was established as a valid instrument for measuring post-surgical linear scars. QoL with scar assessment statistically significantly correlates, 6 months after surgery, with worse scores, particularly notable among women, younger patients, and those with tumours in the cheek region.

## 1. Introduction

Non-melanoma skin cancer (NMSC) remains the most prevalent form of neoplasm, with facial BCC accounting for the majority of NMSC cases [1,2]. The typical approach for treating most BCC is surgery [3]. Despite advanced methods, surgical interventions have not achieved complete scarless healing.

Facial scars tend to elicit the greatest distress among patients, frequently leading to negative psychosocial effects [4,5,6]. Patient-reported outcomes (PROs) are significant measures for evaluating the effectiveness of skin cancer treatment [7]. Various scar characteristics, including the location, morphology [8], and interaction with facial features [9] contribute to its perception. Despite prior efforts to examine scar assessment within exact facial areas, different surgery types [10], and impact on QoL [11], the intricate relationships among these factors remain unclear.

The Patient and Observer Scar Assessment Scale (POSAS 2.0) is a specialised outcome measure instrument for the evaluation of both burn and surgical scars [12,13]. It comprises two distinct parts including the Patient Scar Assessment Scale (PSAS) and the Observer Scar Assessment Scale (OSAS). The PSAS is used to evaluate the perspective of patients and the OSAS is similarly used for professionals. This dual approach allows for comparative analysis regarding opinions on scar quality. The scale has demonstrated superior effectiveness compared with commonly employed measures like the Vancouver Scar Scale (VSS) or the Manchester Scar Scale (MSS). Both the patient and observer scales of the POSAS have been shown to possess greater reliability, objectivity, consistency, and comprehensiveness in evaluating linear scars [12,14,15,16]. To this day, the POSAS has been translated and validated in multiple languages [12,14,17,18,19,20,21,22]. To utilise this instrument for Lithuanian patients, it is imperative to assess its validity within the Lithuanian patient cohort.

The objectives of this study were to (1) translate, culturally adapt, and validate the Lithuanian version of the POSAS 2.0, subsequently utilising this questionnaire in the secondary pilot phase to achieve the following: (2) identify the differences in scar assessment within specific aesthetic facial regions, age, gender, and tumour size groups, and surgery types, and (3) establish empirical correlations between PSAS and short- as well as long-term QoL by employing a prospective-longitudinal study design.

## 2. Patients and Methods

### 2.1. Procedures and Ethics Statement

Permission to translate and validate the POSAS 2.0 into the Lithuanian language was granted by the scale developers in 2022. This study was carried out under the approval of the Vilnius Regional Biomedical Research Ethics Committee (Approval No. 2022/11-1476-943, issued 18th November 2022). In alignment with the Declaration of Helsinki, all study participants provided written informed consent.

Data were collected from 23 November 2022 to 22 November 2023 at the Vilnius University Hospital Santaros Klinikos Centre of Dermatovenereology (VUH), Lithuania.

### 2.2. Patients

Following the recommended sample size for statistical patient-reported outcome measure (PROM) analysis [23,24], a total of 100 consecutive patients were included in this study. Patients with highly suspected or histopathologically confirmed facial BCC diagnosis underwent surgical skin cancer treatment in accordance with the European Association of Dermato-Oncology (EADO), National Comprehensive Cancer Network (NCCN) treatment guidelines, and standardised VUL treatment protocols, resulting in linear postoperative scars. The data were collected at 3 specific time points as follows: 1st visit corresponding to the day of surgery, 2nd visit—1 month post-surgery, and 3rd visit—6 months post-surgery.

Patients who had developed any facial scars as a result of surgical treatment 1 year prior to enrolment, individuals with significant cognitive dysfunction, and those lacking a deep comprehension of the Lithuanian language were excluded from this study.

During the 1st visit, information on socio-demographic and clinical characteristics was gathered. Demographic factors included age, gender, marital status, education, place of residence, employment status, and the presence and frequency of interactions with family members. Clinical details encompassed tumour size, precise tumour location, and the type of surgery performed. The presurgical tumour location was classified into specific regions based on the Facial Aesthetic unit Classification proposed by TT Fattahi [25]. Patients were subsequently grouped into three categories based on the surgery type as follows: (E) excision, (P) skin plasty reconstruction by local flaps, and (T) skin graft transplantation. The participants were provided with paper-based or digital Skin Cancer Index (SCI) and the Dermatology Life Quality Index (DLQI).

At the 2nd and 3rd visits, the patients repeated the SCI and DLQI scales and were asked to evaluate their scars with the PSAS. The same scar was additionally assessed by two observers (AK, DS), a medical student and a plastic surgeon.

### 2.3. Administered Outcome Measures

#### 2.3.1. The Patient and Observer Scar Assessment Scale

The POSAS is used to assess scar quality. A scar is evaluated from 2 perspectives including the patient’s (PSAS) and the observer’s (OSAS).

The PSAS consists of 6 parameters (pain, itchiness, colour, stiffness, thickness, and irregularity). The OSAS consists of 6 parameters (vascularity, pigmentation, thickness, relief, pliability, and surface area). Each parameter is assessed by comparing the scar to the surrounding skin. The score ranges from 1 to 10, where 1 is normal skin and 10 is the worst imaginable scar or sensation. The total score is calculated by summing the scores, with 60 being equivalent to the worst imaginable scar and 6 to normal skin. An additional 7th question concerns the patient’s or observer’s (PSAS and OSAS respectively) overall opinion about the scar (1—normal skin, 10—worst imaginable scar). The score of the Q7 is not added to the total but can be considered as a separate parameter.

#### 2.3.2. The Skin Cancer Index

The SCI is a skin cancer-specific PROM instrument with a focus on emotional, social, and appearance aspects. It consists of 15 Likert scale questions, with scores ranging from 1 (very much—indicating a significant impact on QoL) to 5 (not at all—suggesting no impact on QoL). The total score falls within the range of 15 to 75 points, with a higher score signifying an improved QoL.

#### 2.3.3. The Dermatology Life Quality Index

The DLQI assesses the impact of a skin condition on the patient’s life in the past week. It consists of 10 questions, each rated from 0 to 3. The cumulative score, ranging from 0 to 30, reflects the overall impact on QoL. A higher score signifies an increased impact of the skin problem on the patient’s life, leading to accordingly poorer QoL.

### 2.4. Validation Phase

#### 2.4.1. Translation and Cultural Adaptation

Forward translation.

The POSAS 2.0 underwent translation from English to Lithuanian, following ISPOR TCA [26] guidelines and the COSMIN Study Design Checklist [23,24]. Collaboration among medical staff proficient in both languages ensured accuracy. The forward translation was conducted by an experienced plastic surgeon, followed by a review by a Lithuanian group comprising resident doctors, nurses, and dermatovenereologists. A consensus was reached on the initial Lithuanian PSAS and OSAS versions.
Backward translation.

To verify accuracy, the POSAS 2.0 was backtranslated into English by two independent dermatologists unaware of the original English questionnaire version. Minor linguistic adjustments were made after comparing the back translations with the original text.
Testing.

Cognitive debriefing involved 15 patients with linear facial scars and 15 staff members at VUH. They were asked about comprehension, potential misinterpretations, and relevance of each scale item in scar assessment.
Finalisation.

After reviewing cognitive debriefing results, a consensus was reached on the final version of the POSAS 2.0 in the Lithuanian language.

#### 2.4.2. Statistical Analysis

The statistical analysis was performed using R Statistical Software (version 4.2.2; R Foundation for Statistical Computing, Vienna, Austria) and MedCalc Software Ltd. (version 20.305, Ostend, Belgium; accessed on 1st April 2024). The existence of floor/ceiling effects was acknowledged when >15% of subjects scored at the lowest or highest extremes. A *p* < 0.05 was considered as statistically significant.
Internal consistency.

Cronbach’s alpha coefficient was calculated for PSAS and OSAS at 2 time points. Coefficient values between 0.70 and 0.95 were considered to be adequate [27,28].
Structural validity.

As the POSAS operates on a reflective model, a one-factor confirmatory factor analysis (CFA) was conducted. The criteria for a satisfactory CFA fit were as follows: comparative fit index (CFI) > 0.90 adequate and >0.95 good; Tucker Lewis Index (TLI) (>0.90 adequate and >0.95 good; Root Mean Square Error of Approximation (RMSEA) < 0.08; Standardised Root Mean Squared Residual (SRMR) < 0.08; and chi-squared (χ^2^)/degrees of freedom (df) with the desired range of 2–5 [29].
Construct validity.

Spearman’s rank correlation coefficient (ρ) was calculated for each PSAS/OSAS question in relation to the total score. The resulting coefficient values were interpreted as very strong (0.80–1), strong (0.6–0.799), medium (0.4–0.599), weak (0.2–0.399), and very weak (0–0.199).
Criterion validity.

The PSAS’s convergent validity was assessed using the DLQI questionnaire because of the absence of comparable instruments in the Lithuanian language for wounds or scars. Linguistically, only the first question (Q1) of the DLQI directly relates to the skin discomfort (pain and itching). Spearman’s rank correlation coefficient was employed to analyse the relationship between PSAS and DLQI after the 2nd visit (PSAS-II and DLQI-II). The following three hypotheses were predefined, and construct validity was considered acceptable if all of them (>75%) were validated [30]:Positive correlation between PSAS-II and DLQI-II overall scores.Positive correlation between PSAS-II-Q1 and DLQI-II-Q1.Positive correlation between PSAS-II-Q2 and DLQI-II-Q2.

The convergent construct validity of OSAS could not be evaluated because there are no other scar evaluation instruments for observers validated in the Lithuanian language.
Measurement error and reliability.

PSAS: A subgroup of 50 patients completed the questionnaire twice within 5–7 days. Paper or digital PSAS questionnaires with identical instructions were provided to patients during both the initial and second administration. The first completion of the PSAS occurred at home, while the second took place in the hospital (during the 2nd or 3rd visit) as the only distinguishing factor. Additionally, the subgroup was asked about the potential factors or changes that could influence answers during the interim period.

OSAS: The scars of 100 study participants were photographed during the 2nd or 3rd visit. The photographs were taken with a Canon EOS 600D, its settings being automatically adjusted to the lightning. The photographs were then reanalysed 1 week after the initial in-person OSAS evaluation by the same observers (AK, DS). This analysis did not incorporate the pliability (Q5) parameter because it could not be evaluated in the photographs.

Inter-tester as well as intra-tester reliability was evaluated by calculating the Intraclass Correlation Coefficient (ICC) using a two-way mixed effects model with absolute agreement (95% CI). ICC values exceeding 0.70 were considered acceptable [30]. The standard deviation of differences (SDdif) was computed to assess the dispersion of differences between test and retest (TR) scores, where a smaller SDdif suggested good agreement between TR scores. The standard error of measurement (SEM) was determined using the following formula: SEM = SDdif√(1−ICC). The smallest detectable change in an individual (SDCind) was calculated as follows: SDCind = 1.96 × √2 × SEM. SDCgroup was derived by dividing SDCind by √n, where n represents the sample size. The mean of the differences between test and retest scores was computed as the mean difference score (MD). Limits of Agreement (LoA) were calculated using the following formula: MD ± 1.96 × SDdiff.
Responsiveness.

The standardised response mean (SRM) was calculated between the 2nd and 3rd visits. It was hypothesised that both the PSAS and the OSAS would show a significant decrease in scores when comparing short- and long-term postoperative results, indicating its responsiveness to healing-induced variations. The null hypothesis assumed no significant difference, while the alternative hypothesis predicted a meaningful change in scores, affirming the questionnaire’s sensitivity to the effects of scar changes.

The questionnaire’s responsiveness was evaluated by conducting statistical comparisons, including *t*-tests and analysis of variance (ANOVA) followed by post hoc tests, to compare scores across various groups.

### 2.5. Pilot Phase

#### 2.5.1. PSAS Score Correlations with QoL

Preliminary correlations between disease-specific QoL and scar assessments at the 2nd and 3rd visits were evaluated using Spearman’s correlation coefficient.

#### 2.5.2. Segment Analysis and the POSAS Score Differences across Anatomic Units

Potential differences in the POSAS scores based on age, gender, tumour size, aesthetic facial units, and surgery groups (E, P, T) were examined at the 2nd and 3rd visits. Student’s *t*-test was employed for binary variables, while ANOVA was utilised for categorical variables with three or more groups. Post hoc tests were conducted to identify specific groups with significantly different means whenever ANOVA yielded statistical significance.

## 3. Results

### 3.1. Descriptive Statistics

In total 100 consecutive patients were included in this study. The PSAS, SCI, and DLQI questionnaires were completed by all study participants during the second and third postoperative visit, along with the OSAS, which was filled out by the observers. One hundred patients for OSAS and a subgroup of fifty for PSAS were reassessed for measurement error calculations. Only one missing value was detected. It was replaced by applying the Mode Imputation method. The instances of floor/ceiling effects were observed as follows: the SCI-II exhibited a negative floor/ceiling effect; the DLQI-II displayed a positive floor effect, with 28 patients achieving the minimum (28% > 15%), while the ceiling effect was negative; the PSAS II showed a negative floor/ceiling effect; and the PSAS III indicated a positive floor effect (23% > 15%), yet the ceiling effect was negative. Both AK and DS in the OSAS II/III exhibited negative floor and ceiling effects. Demographic and clinical information are presented in Table 1 and Table 2.

### 3.2. Translation and Cultural Adaptation

Creating the Lithuanian version of the POSAS 2.0 involved a sequence of steps, including forward translation, backward translation, and a cognitive debriefing process. Together, these methods ensured linguistic precision and cultural relevance, ultimately affirming the face and content validity of the PSAS and OSAS.

### 3.3. Internal Consistency

Cronbach’s alpha values were found to be highly acceptable for both the OSAS-II/III and PSAS-II/III (Table 3). The findings indicate robust internal consistency among the questionnaire items, confirming the instrument’s reliability in evaluating scars. It enhances the reliability of the gathered data for future analyses and interpretation within our study.

### 3.4. Structural Validity

The confirmatory factor analysis for both the OSAS and PSAS confirmed that there is only one main factor for the scales. The modification indices could not suggest any modification that would improve the model results. These results are supported theoretically as questionnaires do not have any subscales. Based on the fit results (CFI, TLI, and SRMR), the model falls within the range of acceptable to good. Nevertheless, there is potential for improvement in reducing the RMSEA (Figure 1 and Figure 2, Table 4).

### 3.5. Construct Validity

The Spearman’s correlations between individual PSAS questions and the total score are displayed in Table 5. The values vary from 0.324 to 0.836, suggesting a weak to very strong positive association. Notably, the weakest correlation was found with Q1 and Q2. However, the remaining correlations surpass 0.8, indicating a very strong coherence among these factors.

Table 6 presents the results for the OSAS with correlations ranging from 0.733 to 0.813. These values signify strong to very strong positive correlations.

### 3.6. Criterion validity

The results reveal a medium positive correlation between the overall scores of PSAS-II and DLQI-II. The strongest correlation was identified between PSAS-II-Q1, Q2 and DLQI-II-Q1, highlighting the questions’ focus on symptom evaluation (Table 7).

### 3.7. Measurement Error and Reliability

#### 3.7.1. PSAS

Fifty patients underwent scar reassessment to evaluate the test–retest reliability of the PSAS. The ICC value obtained for the total score was 0.729 (95% CI = 0.568–0.837). Most ICC values for single questions exceeded the threshold of 0.7, with the exceptions being PSAS-I-Q5 and PSAS-I-Q6, which pertain to scar thickness and irregularity, respectively (Table 8).

#### 3.7.2. OSAS

One hundred patients’ scars were re-evaluated by two observers (AK, DS) for the assessment of intra- and inter-tester reliability of the OSAS. The findings reveal that the ICC values for AK are below 0.7, indicating poor consistency and reliability between AK’s observations. However, the results demonstrate that the ICC values for the DS observer were satisfactory for all questions, exceeding 0.7 and indicating reliable consistency between DS’s observations (Table 9).

In the comparison of AK’s and DS’s assessments, the observers generally show consistent agreement on vascularity (Q1) and overall opinion (Q7) parameters. Additionally, there is acceptable consistency in the overall scores provided by both observers. Across the first and second evaluations, the ICC values between DS and AK tend to hover around borderline acceptability, indicating minor discrepancies between the observers. This suggests that one observer might rate a scar as worse for one aspect and better for another, yet ultimately resulting in a final score that is reliably consistent between both evaluators (Table 10).

### 3.8. Responsiveness and Agreement between the PSAS and OSAS

When comparing the scores between the second and third visit, statistically significant changes were observed in both the PSAS (*p* < 0.001, mean difference −8.44 points) and the OSAS (*p* < 0.001, mean difference −8.18 points). The results indicate significant improvement in scar evaluation both by patients and observers, emphasizing the profound impact time has on scar healing and its eventual assessment (Table 11).

Large responsiveness levels for both the PSAS and OSAS were observed when comparing the scores of the second and third visits (SRM > ± 0.8). The results confirm that the Lithuanian POSAS effectively detects changes in scars over time. The patient and observer ratings of scars significantly correlated both during the second and third postoperative visits. A low correlation between the PSAS and OSAS scores was observed 1 month after surgery and a medium correlation 6 months after surgery.

### 3.9. Correlations between the Scar Assessment and QoL

One month post-surgery, the PSAS scores showed no significant correlation (*p* > 0.05) with the SCI, indicating a lack of association between scar assessment of QoL at this early stage of recovery. Nevertheless, 6 months after surgery, the connections between scar assessment and QoL became apparent. The findings revealed medium negative correlations with SCI Total and its components (Social, Emotional, and Appearance subscales). This suggests that as PSAS scores increase, SCI scores decrease, signifying poorer QoL for patients who perceive their scars more negatively (Table 12).

### 3.10. Segment Analysis and the POSAS Score Differences across Anatomic Units

To ensure the ability to use the statistical tests during the pilot phase, the examination of score distributions across anatomical units did not include finer subunits, as there were not enough cases. Table 13 presents the segment analysis for the PSAS and OSAS scores. Because the following groups consisted of one patient, they were excluded from the statistical analysis:Eyelid 80–91 age group, *n* = 1;Upper lip 57–69 age group, *n* = 1;Eyelid T group, *n* = 1;Upper lip P group −1, *n* = 1;Upper lip T group, *n* = 1;Nose > 15 mm tumour group, *n* = 1;Eyelid > 15 mm tumour group, *n* = 1;Upper lip > 15 mm tumour group, *n* = 1.

#### 3.10.1. Scar Assessment 1 Month Post-Surgery

PSAS-II: Statistically significant differences are evident in scar assessment among the age groups in the forehead anatomic unit. Post hoc analysis revealed that the 70–79 year group rates forehead scars statistically significantly worse than the 80–90 year group.

OSAS-II: Statistically significant differences emerge in nasal scar assessments by observers for patients in the 34–56 year group vs. the 80–91 year group. The results suggest that the observers rated nasal scars statistically significantly worse for the patients in the 34–56 year group. After surgical excision of larger tumours (6–10 mm), the observers rated the scars worse in the eyelid region than those that were smaller (≤5 mm). This suggests that larger tumours result in bigger scars, which become an influencing factor in their evaluation. These findings underscore the influence of age and size on scar perception and the nuanced assessments made by observers across the different anatomical regions.

#### 3.10.2. Scar Assessment 6 Months Post-Surgery

PSAS-III: Gender disparities in scar assessment became apparent, with women consistently rating scars in the late postoperative phase significantly worse than men. When considering the specific anatomical units, the discrepancies were most pronounced and statistically significant in the cheek region. Furthermore, variations among age groups were noticed, notably with the 34–56 age group evaluating scars significantly worse than both the 70–79 and 80–91 age groups. These distinctions were statistically significant in the cheek and upper lip anatomical regions.

OSAS-III: Observers rated scars statistically significantly worse for men in the forehead region.

## 4. Discussion

The first phase of this study resulted in the successful translation and validation of the Lithuanian version of the POSAS 2.0 for linear scars. A comprehensive assessment of the psychometric properties of both the PSAS and OSAS demonstrated that they exceeded acceptable thresholds for internal consistency, structural validity, criterion validity, construct validity, reliability, and responsiveness. Both PSAS 2.0 and OSAS 2.0 demonstrated strong internal consistency values during the second and third visits (Cronbach’s alpha > 0.7).

The floor effect was observed to be present in PSAS-III, with 23% of patients attaining the minimum scores. These results are consistent with those of a Finnish validation study [20], which linked the floor effect to the evolving dynamics following the acute healing phase post-surgery. CFA findings confirmed the scale analysis by van de Kar et al. [12], showing that both scales comprised a single factor. Most PSAS 2.0 questions showed significant alignment with the scale’s intended construct. However, Q1 and Q2, concerning itching and colour, respectively, did not exhibit a strong correlation with the overall score. The reason related to linguistic phrasing was ruled out as patients reported no comprehension difficulties. It is plausible that these questions tap into different facets of the construct that are not adequately reflected in the total scale score, making them comparatively less relevant than other questions. From the observer’s perspective, all questions showed robust correlations with the overall score. This suggests that each question of OSAS 2.0 contributes meaningfully to the scar assessment by the observers.

Because of limited PROM resources in the Lithuanian language, only the PSAS was assessed for criterion validity, establishing the correlations with the DLQI. Statistically significant correlations were found between both questionnaires, confirming, that the scale accurately captures the characteristics of the symptoms related to skin discomfort.

The results suggest that PSAS generally maintains reliability in repeated measurements over time, as most questions exhibit strong consistency between assessments. However, challenges with scar thickness (Q5) and irregularity (Q6) assessments underscore potential areas for enhancing the scale’s reliability. In contrast to the calculations for the Norwegian OSAS conducted by Hjellestad et al. [19], only one evaluator achieved acceptable intra-observer reliability scores for the Lithuanian version. Similar discrepancies were reported for the Italian version of the OSAS [31]. Moreover, these differences may stem from using photographic evaluations. While some studies confirm photographic equivalence [32], others, including the scale authors themselves [12], suggest the POSAS cannot be accurately assessed via photographs. Given that many patients lived far away, asking for additional in-office scar evaluations was impractical, justifying the use of photographic assessments in clinical practice. The less consistent intra-tester results for OSAS 2.0 emphasise the necessity for further investigation to ensure dependable and consistent observations from the same rater.

Confirming the findings of the existing research [33], we identified disparities in OSAS 2.0 scores among different observers. The inter-tester calculations suggest that while evaluators rate the scars differently by a single parameter, they tend to agree on the criteria for vascularity (Q1), overall opinion (Q7), and the total score, suggesting that these parameters are less subjective and more reliably interpreted across different raters.

Although the POSAS has been shown to demonstrate good reliability in evaluating various scar types (e.g., zigzag, circular, burn, linear) [34,35], we observed a lack of criteria for assessing lymphostasis, which may arise from the lymph-disrupting nature of scars, particularly those on the face. This issue is especially prominent in skin plasty (P) and transplantation (T) groups, where the surgical procedure itself poses a heightened risk of such complications. This phenomenon was observed when patients displayed satisfactory single scar characteristics, yet experienced significant facial disfigurement because of lymphostasis, which was reflected in their overall PSAS scores. The POSAS currently lacks criteria for evaluating such instances because parameters like relief, thickness, and surface area primarily pertain to the scar itself rather than the surrounding tissues. In addition to that, a small percentage of our cohort expressed confusion regarding their overall scar assessments, graded on a scale from 1 to 10, where 10 indicated the poorest scar quality. This confusion might stem from the evaluation practices in Baltic countries, where higher scores typically signify superior quality. This discrepancy was not mentioned in any previous validation studies.

Our study results validate the effectiveness of the Lithuanian POSAS 2.0 in detecting changes in scars over time and highlight the statistically significant improvement seen between the short and late postoperative periods. Patient and observer evaluations show a statistically significant correlation in scar quality at two different time points, further supporting the reliability of the PSAS and OSAS.

Although differences in POSAS scores across anatomical regions were highly expected, statistically significant variations in scar assessment were only evident when patients were grouped by gender, age, surgery type, and tumour size. During the early postoperative period, it was noted that relatively younger patients evaluated scars on their foreheads more critically. However, for this specific group, the observers focused their attention more on the nose anatomic region. For the patients who had larger primary tumours and, consequently, longer scars, the observers distinguished the eyelid anatomic unit within which the scars were evaluated worse. Six months post-surgery, statistically significant gender disparities became apparent, with several anatomical units, such as the cheek and upper lip, predicting worse scores for younger women. Conversely, observers reported worse scars for men on the forehead. Despite limited feasibility for post hoc tests, identifying significant distinctions laid the groundwork for future research with a larger sample size.

In our prior examination of QoL among patients with facial BCC, we noted a statistically significant improvement at the 6-month mark following surgery. In this study, we investigated whether scar quality might be linked to QoL during both the early and late postoperative phases. Significant correlations were identified between PSAS and SCI scores, particularly with the Appearance subscale, at the 6-month post-surgery mark. This indicates a direct association between scar appearance and patient QoL during the later stages of recovery. These findings offer valuable insights into how patients perceive the aesthetic aspects of scars following skin cancer surgery and its impact on their overall QoL. Preliminary findings from post hoc analysis provide a basis for future investigations, including the addition of advanced statistical techniques such as linear regression and the consideration of various factors like anthropometric variables and socio-demographic characteristics.

This study is one of the few to evaluate the psychometric properties of both PSAS 2.0 and OSAS 2.0, following a rigorous guideline-based methodology by COSMIN. The additional pilot phase of this study represents the first analysis of the intricate connections between post-interventional patient satisfaction with aesthetic outcomes in specific facial anatomical regions and its correlation with disease-specific QoL, utilizing a prospective longitudinal study design. This allowed for the refinement of research protocols and assessment tools, ultimately improving the quality and efficiency of future investigations on this topic.

Acknowledging its limitations, the pilot phase of this study had an insufficient sample size for complete factor analysis, which may limit the findings that can be applied to broader populations. Moreover, this study’s focus on the Lithuanian patient population may limit the applicability of results to cultural contexts beyond the Baltic region. These limitations underscore the need for ongoing longitudinal study and suggest incorporating strategies to overcome them.

## 5. Conclusions

The Lithuanian version of the POSAS 2.0 can be confidently used for assessing scar quality in both clinical and research settings, offering comprehensive insights from both patient and observer perspectives. Notably, there is a statistically significant improvement in scar quality observed 6 months post-surgery, correlating with enhanced QoL. Analysis of PSAS scores revealed certain demographic groups, particularly younger women, which tend to evaluate scars more critically. Additionally, specific facial areas—forehead, upper lip, and cheek—were identified as aesthetically sensitive. Conversely, observers show sensitivity towards the male gender and their scars on the forehead, nose, and eyelid, with larger presurgical tumour size correlating with poorer OSAS scores.

## Figures and Tables

**Figure 1 cancers-16-02091-f001:**
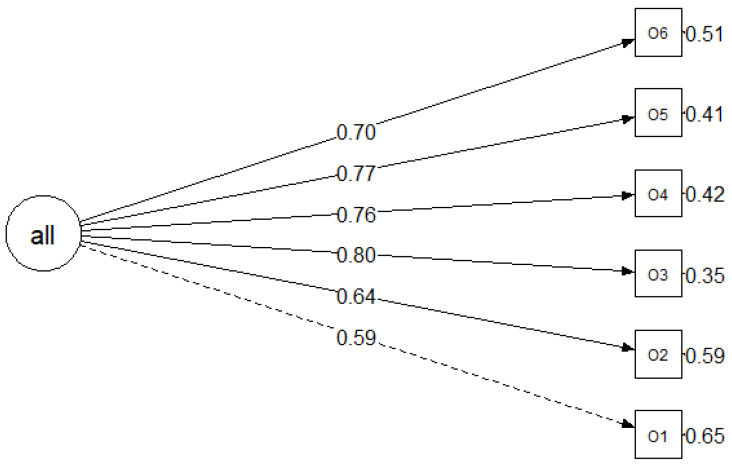
Graphical representation of the refined model (Model 1), with standardised values.

**Figure 2 cancers-16-02091-f002:**
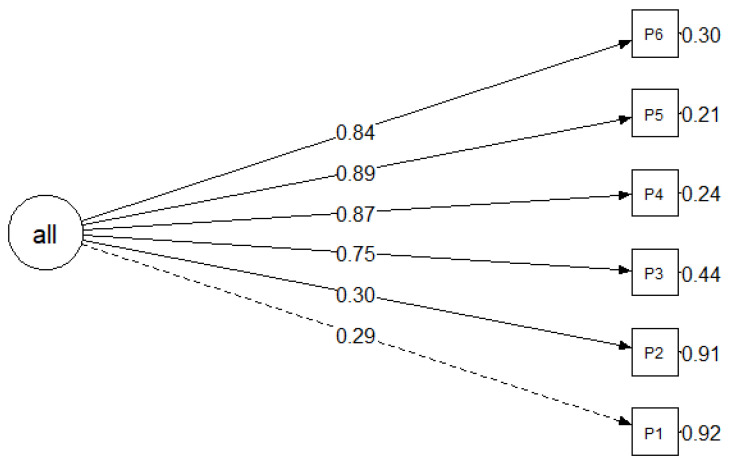
Graphical representation of the refined model (Model 2), with standardised values.

**Table 1 cancers-16-02091-t001:** Patient sociodemographic characteristics.

Sociodemographic Characteristics, *n* = 100	
Age	68.31 ± 12.979
34–56 years	20
57–69 years	27
70–79 years	32
80–91 years	21
Gender, *n* (%)	
Female	72 (72%)
Male	28 (28%)
Marital status, *n* (%)	
Divorced	9 (9%)
Widow/widower	22 (22%)
Living together with a partner	4 (4%)
Dating but living separately	1 (1%)
Married	59 (59%)
Single	5 (5%)
Education, *n* (%)	
Non-university higher education	22 (22%)
Other (professional schools)	8 (8%)
Basic (8–10 grades)	5 (5%)
Primary	2 (2%)
University degree	51 (51%)
Secondary (11–12 grades)	12 (12%)
Residence, *n* (%)	
Village (<500 inhabitants)	4 (4%)
City (>3000 inhabitants)	90 (90%)
Town (500–3000 inhabitants)	6 (6%)
Employment, *n* (%)	
Employed	31 (31%)
Employed but retired	1 (1%)
Employed (home office)	2 (2%)
Unemployed	6 (6%)
Retired	60 (60%)
Do you have children/close relatives? *n* (%)	
No	4 (4%)
Yes	96 (96%)
Do you often meet them? *n* (%)	
No	7 (7%)
Yes	93 (93%)

**Table 2 cancers-16-02091-t002:** Patient clinical characteristics.

Clinical Characteristics, *n* = 100	
Largest tumour diameter, mm	9.44 ± (4.948); range: 3–30
Tumour size group, *n* (%)	
0–5 mm	16 (16%)
6–10 mm	57 (57%)
11–15 mm	16 (16%)
>15 mm	11 (11%)
Tumour location by TT Fattahi, *n* (%)	
1—Forehead unit	32 (32%)
1a—central subunit	14
1b—lateral subunit	18
2—Nasal unit	26 (26%)
2.1—tip subunit	8
2.3,6—right and left alar base subunits	9
2.4,5—right and left alar side wall subunits	4
2.7—dorsal subunit	5
2.8,9—right and left dorsal side wall subunits	5
3—Eyelid unit	8 (8%)
3a—lower lid subunit	4
3b—upper lid subunit	1
3c—lateral canthal subunit	1
3d—medial canthal subunit	2
4—Cheek unit	28 (28%)
4a—medial subunit	14
4b—zygomatic subunit	3
4c—lateral subunit	4
4d—buccal subunit	7
5—Upper lip unit	3 (3%)
5b—lateral subunit	3
Surgery groups, *n* (%)	
E	49 (49%)
P	38 (38%)
T	13 (13%)

**Table 3 cancers-16-02091-t003:** Cronbach’s alpha values for the OSAS and PSAS questionnaires at the 2nd and 3rd visit.

	II	III
OSAS, AK	0.855	0.822
OSAS, DS	0.845	0.793
PSAS	0.828	0.836

**Table 4 cancers-16-02091-t004:** Fit results of the models tested (*n* = 100).

	Description	χ^2^	df	RMSEA (95% CI)	CFI	TLI	SRMR
Model 1	The original model of the OSAS questionnaire with one factor	19.691	9	0.109 (0.019; 0.186)	0.956	0.926	0.052
Model 2	The original model of the PSAS questionnaire with one factor	21.201	9	0.116 (0.035; 0.193)	0.956	0.927	0.049

**Table 5 cancers-16-02091-t005:** Spearman’s correlation coefficients between PSAS scores.

	P1	P2	P3	P4	P5	P6	P7
PSAS-II Total score	0.324 **	0.377 ***	0.807 ***	0.814 ***	0.841 ***	0.785 ***	0.836 ***

Significance: **—*p*-value < 0.01, ***—*p*-value < 0.001.

**Table 6 cancers-16-02091-t006:** Spearman’s correlation coefficients between OSAS scores.

	O1	O2	O3	O4	O5	O6	O7
OSAS-II Total score	0.733 ***	0.736 ***	0.754 ***	0.734 ***	0.786 ***	0.756 ***	0.813 ***

Significance: ***—*p*-value < 0.001.

**Table 7 cancers-16-02091-t007:** Spearman’s correlation coefficients between PSAS-II and DLQI-II.

	DLQI-II-Q1	DLQI-II Total Score
PSAS-II-Q1	0.428 ***	0.44 ***
PSAS-II-Q2	0.724 ***	0.583 ***
PSAS-II-Q3	0.253 *	0.36 ***
PSAS-II-Q4	0.156	0.197 *
PSAS-II-Q5	0.129	0.194
PSAS-II-Q6	0.127	0.232 *
PSAS-II-Q7	0.158	0.276 **
PSAS-II Total score	0.358 ***	0.423 ***

Significance: *—*p*-value < 0.05, **—*p*-value < 0.01, ***—*p*-value < 0.001.

**Table 8 cancers-16-02091-t008:** Test–retest reliability of the PSAS.

	Test	Retest	ICC (95% CI)	MD (95% CI)	SDdiff	SEM	SDCind	SDCgroup	95% LoA
Pain PSAS-Q1	2.02 ± 1.363	2.02 ± 1.478	0.721 (0.555; 0.832)	0 (−0.296; 0.296)	1.069	0.565	1.565	0.221	−2.095; 2.095
Itchiness PSAS-Q2	2.54 ± 1.764	2.56 ± 1.74	0.827 (0.713; 0.898)	−0.02 (−0.308; 0.268)	1.04	0.433	1.199	0.17	−2.058; 2.018
Colour PSAS-Q3	3.36 ± 1.758	3.02 ± 1.79	0.74 (0.581; 0.844)	0.34 (−0.08; 0.688)	1.255	0.640	1.774	0.251	−2.12; 2.8
Stiffness PSAS-Q4	3.4 ± 1.906	3.32 ± 1.845	0.72 (0.544; 0.831)	0.08 (−0.311; 0.471)	1.412	0.747	2.071	0.293	−2.687; 2.847
Thickness PSAS-Q5	3.2 ± 1.895	3.02 ± 1.755	0.618 (0.413; 0.764)	0.18 (−0.263; 0.623)	1.6	0.989	2.741	0.388	−2.955; 3.315
Irregularity PSAS-Q6	3.02 ± 1.801	3.1 ± 1.776	0.679 (0.496; 0.805)	−0.08 (−0.479; 0.319)	1.441	0.816	2.263	0.32	−2.903; 2.743
Overall opinion PSAS-Q7	2.88 ± 1.612	2.94 ± 1.621	0.727 (0.564; 0.835)	−0.06 (−0.393; 0.273)	1.202	0.628	1.741	0.246	−2.416; 2.296
Total score	17.54 ± 8.311	17.04 ± 7.918	0.729 (0.568; 0.837)	0.5 (−1.163; 2.163)	6.001	3.124	8.659	1.225	−11.262; 12.262

**Table 9 cancers-16-02091-t009:** Intra-tester reliability of the PSAS.

	Test		Retest	ICC (95% CI)	MD (95% CI)	SDdiff	SEM	SDCind	SDCgroup	95% LoA
Vascularity OSAS-II-Q1	DS	2.59 ± 1.19	2.54 ± 1.039	0.844 (0.766; 0.892)	0.05 (−0.073; 0.173)	0.626	0.247	0.685	0.069	−1.177; 1.277
	AK	2.53 ± 1.201	2.53 ± 0.904	0.627 (0.491; 0.733)	0 (−0.181; 0.181)	0.921	0.562	1.559	0.156	−1.805; 1.805
Pigmentation OSAS-II-Q2	DS	2.26 ± 0.906	2.3 ± 0.759	0.785 (0.696; 0.85)	−0.04 (−0.148; 0.068)	0.549	0.255	0.706	0.071	−1.116; 1.036
	AK	2.82 ± 0.892	2.69 ± 0.734	0.532 (0.376; 0.658)	0.13 (−0.024; 0.284)	0.787	0.538	1.492	0.149	−1.412; 1.672
Thickness OSAS-II-Q3	DS	2.45 ± 1.077	2.32 ± 0.942	0.82 (0.742; 0.876)	0.13 (0.013; 0.247)	0.597	0.253	0.702	0.07	−1.041; 1.301
	AK	2.79 ± 0.769	2.68 ± 0.75	0.457 (0.289; 0.599)	0.11 (−0.045; 0.265)	0.79	0.582	1.614	0.161	−1.438; 1.658
Relief OSAS-II-Q4	DS	2.36 ± 1.142	2.29 ± 0.957	0.743 (0.641; 0.819)	0.07 (−0.078; 0.218)	0.756	0.383	1.062	0.106	−1.411; 1.551
	AK	2.67 ± 0.943	2.51 ± 0.87	0.544 (0.391; 0.668)	0.16 (−0.009; 0.329)	0.861	0.581	1.612	0.161	−1.528; 1.848
Pliability OSAS-II-Q5	-									
Surface Area OSAS-II-Q6	DS	1.82 ± 0.796	1.83 ± 0.766	0.762 (0.665; 0.833)	−0.01 (−0.116; 0.096)	0.541	0.264	0.732	0.073	−1.071; 1.051
	AK	2.66 ± 0.807	2.43 ± 0.868	0.483 (0.315; 0.621)	0.23 (0.066; 0.394)	0.839	0.603	1.672	0.167	−1.415; 1.875
Overall opinion OSAS-II-Q7	DS	2.64 ± 1.02	2.49 ± 0.916	0.805 (0.719; 0.866)	0.15 (0.034; 0.266)	0.592	0.289	0.801	0.08	−1.011; 1.311
	AK	2.96 ± 0.909	2.79 ± 0.856	0.665 (0.537; 0.763)	0.17 (0.031; 0.309)	0.711	0.412	1.141	0.114	−1.224; 1.564
Total score	DS	13.87 ± 5.15	13.4 ± 4.422	0.887 (0.836; 0.923)	0.47 (0.03; 0.91)	2.245	0.755	2.092	0.209	−3.93; 4.87
	AK	13.47 ± 3.597	12.84 ± 2.943	0.61 (0.47; 0.72)	0.63 (0.068; 1.193)	2.87	1.792	4.968	0.497	−4.995; 6.255

**Table 10 cancers-16-02091-t010:** Inter-tester reliability for the 1st and 2nd evaluation.

		Vascularity	Pigmentation	Thickness	Relief	Pliability	Surface Area	Overall Opinion	Total Score
ICC (95% CI)	First evaluation	0.798 (0.699; 0.864)	0.658 (0.216; 0.824)	0.681 (0.487; 0.796)	0.693 (0.529; 0.798)	-	0.43 (−0.254; 0.716)	0.783 (0.631; 0.866)	0.79 (0.688; 0.859)
	Second evaluation	0.769 (0.656; 0.844)	0.608 (0.297; 0.767)	0.547 (0.291; 0.705)	0.622 (0.437; 0.746)	-	0.544 (0.033; 0.757)	0.732 (0.564; 0.83)	0.742 (0.617; 0.826)

**Table 11 cancers-16-02091-t011:** Standardised response mean (SRM) and meaningful changes between the 2nd and 3rd visits.

		*p*-Value	Mean Difference	SRM Value	95% CI	PSAS-II vs. OSAS-II		PSAS-III vs. OSAS-III	
						*p*-Value	Spearman’s Correlation	*p*-Value	Spearman’s Correlation
Second visit vs. third visit	PSAS	0.000	−8.44	−1.06	−1.30 to −0.80	*p* = 0.001	0.32	*p* = 0.000	0.51
	OSAS	0.000	−8.18	−2.10	−2.43 to −1.75				

Significance: *p*-value < 0.05.

**Table 12 cancers-16-02091-t012:** Correlations between the PSAS and SCI.

		PSAS-II	PSAS-III
SCI Total	*p*-value	>0.05	<0.001
	Spearman’s correlation	−0.17	−0.47
SCI Emotional	*p*-value	>0.05	<0.001
	Spearman’s correlation	−0.19	−0.38
SCI Social	*p*-value	>0.05	<0.001
	Spearman’s correlation	0.00	−0.39
SCI Appearance	*p*-value	>0.05	<0.001
	Spearman’s correlation	−0.19	−0.51

Significance: *p*-value < 0.05.

**Table 13 cancers-16-02091-t013:** Segment analysis and the POSAS score differences across anatomic units.

	Second Visit		Third Visit	
	PSAS-II	OSAS-II	PSAS-III	OSAS-III
Anatomic Unit	No, *p* > 0.05	No, *p* > 0.05	No, *p* > 0.05	No, *p* > 0.05
Gender	No, *p* > 0.05	No, *p* > 0.05	Yes, *p* = 0.034	No, *p* > 0.05
- Score differences between men and women by anatomic units.	No	No	Yes	Yes
Age group	No, *p* > 0.05	Yes, *p* = 0.031	Yes, *p* = 0.000	No, *p* > 0.05
- Score differences between age groups by anatomic units.	Yes	Yes	Yes	No
Surgery group (E, P, T)	No, *p* > 0.05	No, *p* > 0.05	No, *p* > 0.05	No, *p* > 0.05
- Score differences between surgery groups by anatomic units.	No	No	No	No
Size group	No, *p* > 0.05	No, *p* > 0.05	No, *p* > 0.05	No, *p* > 0.05
- Score differences between size groups by anatomic units.	No	Yes	No	No

Significance: *p*-value < 0.05.

## Data Availability

Data analysed during this study are not publicly available but are available from the corresponding author upon reasonable request.

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
