# Peer review of "Aesthetic Evaluation of Facial Scars in Patients Undergoing Surgery for Basal Cell Carcinoma: A Prospective Longitudinal Pilot Study and Validation of POSAS 2.0 in the Lithuanian Language"

_cancers, 2024, doi:10.3390/cancers16112091_

Round 1

Reviewer 1 Report

Comments and Suggestions for Authors

The study findings underscore the importance of addressing aesthetic satisfaction in surgically treated basal cell carcinoma patients, given the observed correlations between scar assessment and quality of life. This highlights the need for comprehensive care that considers both the medical and psychosocial aspects of patients' experiences.

The manuscript is very well written, with appropriate paragraph structure and layout.

The tables are correct. I miss some images showing the before and after of the basal cell carcinoma intervention. The references are adequate and up-to-date

1.  The authors propose the validation in the Lithuanian language of an evaluation of residual facial scars after surgical treatment of basal cell carcinoma (POSAS 2.0).
2.The study findings underscore the importance of addressing aesthetic satisfaction in surgically treated basal cell carcinoma patients, given the observed correlations between scar assessment and quality of life.  
3. The main novelty is the validation in the Lithuanian language of the POSAS 2.0 scale, which has not been previously performed in this type of patient.
4. The manuscript is well structured, although minor editing of the English language is necessary, due to minor grammatical errors throughout the manuscript.
5. The conclusions of the manuscript are consistent with the development of the study, as it has internal and external cohesion. It is therefore susceptible to extrapolate the results to other patients.
6. The references are adequate and up to date.
7.  The tables are correct. I miss some images showing the before and after of the basal cell carcinoma intervention.

Comments on the Quality of English Language

The manuscript is well structured, although minor editing of English language is necessary, due to minor grammatical errors throughout the manuscript.

Author Response

Dear Reviewer,

Thank you for your time while reviewing our manuscript. We very much appreciate your insight. Considering your comments, we have edited the manuscript and reflected on your comments, with special attention to the grammar mistakes. The corrections are presented as the “Track Changes” function of MS Word.

Regarding patient pictures of the before and after treatment, due to the sensitive patient identification and medical data we decided against including any identifying data.  

Once again thank you for your review.

Sincerely,

Domantas Stundys - on behalf of all the authors

Reviewer 2 Report

Comments and Suggestions for Authors

1. The manuscript lacks details on experimental replication, which is a significant concern. Please revise to include information on your experimental and technical replications.

2. Enhance the background descriptions for tumors by citing references 10.1039/C7RA01052D and 10.1038/s41563-019-0503-4. Additionally, explain the advantages of the current work compared to related published articles.

3. Strengthen the Abstract section by clearly highlighting the important results and key conclusions, using precise language.

4. The article contains formatting errors, such as incorrect reference spellings that must adhere to the journal's style (e.g., Reference 2). Carefully review and properly use abbreviations.

Author Response

Dear Reviewer,

Thank you for taking the time to review our manuscript. We appreciate your comments and insights.

#1 The clarity was enhanced by further explaining the basal cell carcinoma treatment guidelines we applied during the study and clarifying the time points of the study. Information from Paragraph 2.4 was relocated to Paragraph 2.2. The numbering of the following paragraphs was consecutively corrected.

#2 We found no relevant studies based on the references you provided in the Reviewer’s form. The advantages of our study compared to related articles are presented in greater detail in the Discussion section of the manuscript.

#3 Considering the extensive work we have performed during the study and the achieved results, we struggled to fit it all in the abstract of 200 words. The important results and key conclusions of our study were the successful validation of the translated Lithuanian version of POSAS 2.0 as well as the established correlations between the QoL and scar appearance, especially evident in the later postoperative period.

#4 We have revised the manuscript and the reference list. The formatting was corrected according to the journal’s requirement.

Once again thank you for your review.

Sincerely,

Domantas Stundys - on behalf of all the authors